# iRhom2: An Emerging Adaptor Regulating Immunity and Disease

**DOI:** 10.3390/ijms21186570

**Published:** 2020-09-08

**Authors:** Mazin A. Al-Salihi, Philipp A. Lang

**Affiliations:** Department of Molecular Medicine II, Medical Faculty, Heinrich-Heine-University Düsseldorf, Universitätsstrasse 1, 40225 Düsseldorf, Germany; alsalihi@hhu.de

**Keywords:** iRhom2, *Rhbdf2*, ADAM17, TACE, ectodomain shedding, EGFR, TNF, MAVS, STING

## Abstract

The rhomboid family are evolutionary conserved intramembrane proteases. Their inactive members, iRhom in *Drosophila melanogaster* and iRhom1 and iRhom2 in mammals, lack the catalytic center and are hence labelled “inactive” rhomboid family members. In mammals, both iRhoms are involved in maturation and trafficking of the ubiquitous transmembrane protease a disintegrin and metalloprotease (ADAM) 17, which through cleaving many biologically active molecules has a critical role in tumor necrosis factor alpha (TNFα), epidermal growth factor receptor (EGFR), interleukin-6 (IL-6) and Notch signaling. Accordingly, with iRhom2 having a profound influence on ADAM17 activation and substrate specificity it regulates these signaling pathways. Moreover, iRhom2 has a role in the innate immune response to both RNA and DNA viruses and in regulation of keratin subtype expression in wound healing and cancer. Here we review the role of iRhom2 in immunity and disease, both dependent and independent of its regulation of ADAM17.

## 1. Introduction

Rhomboids are an evolutionarily conserved family of multi-span transmembrane proteins [1]. Some are catalytically active serine proteases capable of intramembrane cleavage of their substrates. These were originally discovered in *Drosophila melanogaster* as intramembrane proteases of EGFR ligands [2,3,4,5]. Other rhomboid family members, while structurally similar, are catalytically inactive and known as pseudoproteases. These rhomboid pseudoproteases were nonetheless found to be evolutionarily conserved indicating the presence of selective pressure despite absence of their proteolytic activity [6]. This alludes to functions beyond rhomboid catalytic activity that are important enough to be preserved. iRhoms 1 and 2, encoded by the genes *Rhbdf 1* and *2*, belong to the latter family of pseudoproteases. They were named iRhoms to indicate their proteolytic inactivity while being part of the rhomboid family [7].

Both *Drosophila* and mammalian iRhoms bind their partner proteins in the ER. *Drosophila*’s single iRhom interacts with EGF family ligands resulting in ER associated ligand degradation (ERAD) and inhibition of EGFR signaling [8]. However, the two mammalian iRhoms bind to ADAM17 rather than the EGFR ligands, which results in trafficking of ADAM17 to the Golgi where it can begin its post-translational modification journey to become a cell surface active protease. Furthermore, iRhom2 is reported to be involved in immune responses seemingly unrelated to ADAM17 activity [9,10]. There are excellent reviews on iRhom and ADAM17 structure [11], functional regulation and expression patterns [12], cellular and pathophysiological roles as well as novel binding partners [13] and combined biological functions [14]. This review however will focus on iRhom2, both as a regulator of ADAM17 as well as its ADAM17-independent roles in immunity and disease.

## 2. Regulation of ADAM17 by iRhom2

Understanding the journey of ADAM17 from production to function is central to understanding how iRhoms modulate ADAM17 activity. Ectodomain shedding of cell surface transmembrane proteins plays a central role in several signaling pathways such as the EGFR and TNF pathways. These cleavage events are performed by metalloproteases, such as ADAM17, also known as TNFα converting enzyme (TACE) in reference to the original discovery of its role in TNFα shedding [15,16,17]. ADAM17 can shed a variety of ligands including the adhesion molecule L-selectin, transforming growth factor alpha (TGFα), the IL-6 receptor (IL-6R) as well as the TNFα receptors (TNFR) 1 and 2 themselves [18,19]. These and a list of over 80 biologically active molecules shed by ADAM17, including growth factors, cytokines, receptors and adhesion molecules are involved in a wide array of physiological processes and diseases from innate immune responses to carcinogenesis [20]. Therefore, it is not surprising that ADAM17 is regulated on multiple levels as briefly summarized here to highlight the interactions with iRhom2 (Figure 1), while a detailed description of these mechanisms have been reviewed elsewhere [12,20].

Once produced, ADAM17 requires several post-translational modifications to become catalytically active. It is proteolytically processed in the Golgi by the proprotein-convertases PC7 and furin to remove its inhibitory pro-domain [21,22,23]. Its cytoplasmic tail is phosphorylated by extracellular signal-regulated kinases (ERK) and mitogen activated protein kinases (MAPK) as part of protein kinase C (PKC) signaling activity, as well as by Polio like kinase 2 (PLK2) [24,25,26,27,28,29,30,31]. Phosphorylation of ADAM17′s cytoplasmic tail, or its deletion, is required for ADAM17 proteolytic activity as well as its trafficking to the cell surface [32]. Unless phosphorylated, the tail serves to dimerize ADAM17 which in turn allows tissue inhibitor of metalloprotease 3 (TIMP3) to bind and inhibit ADAM17 activity. Therefore, deletion or phosphorylation of the cytoplasmic tail allows ADAM17 to become catalytically active [33,34,35,36]. Moreover, ADAM17 substrate phosphorylation has also been reported to influence its ability to shed these post-translationally modified substrates [28,37]. At the cell surface, alteration in the intramolecular disulfide bridge status of ADAM17 by disulfide isomerases, or in response to changes in the local redox environment such as during inflammatory events, induces a conformational change regulating ADAM17 activity as well [38,39,40,41,42]. The ADAM17-iRhom1 and iRhom2 interaction governs ADAM17′s maturation, trafficking towards the cell surface and even its proteolytic activity and substrate selectivity [43,44,45,46,47,48,49]. While ADAM17 has been reported to stabilize cell surface iRhom2 in return [50]. Similar to ADAM17 iRhom2 levels are regulated transcriptionally. iRhom2 is a p63 transcription factor target gene in keratinocytes [51], is induced in response to exposure to lipopolysaccharides (LPS) or viral infections in macrophages [9,43,44] and induced in the liver post bile duct ligation [52]. It may also be regulated through proteasomal mediated degradation. Its half-life can be extended by treating cells with the proteasomal inhibitor MG-132 [53]. However, direct evidence of k48 ubiquitination and proteasomal degradation is still lacking.

Once produced both iRhom1 and iRhom2 bind directly to ADAM17 [49]. The ADAM17-iRhom2 interaction was originally elucidated in relation to ADAM17 shedding of TNFα in hematopoietic cells in iRhom2 deficient mice [43,44]. This interaction was shown to be specific to ADAM17 and not its closest relative ADAM10 [43]. A panel of other ADAM proteins were also tracked in double iRhom 1 & 2 knock out fibroblasts; none of which were found to traffic differently in the absence of iRhoms [49]. The iRhom2 loss of function sincere (*sin*) mutation provided more nuanced insights into the complete mechanism of iRhom-ADAM17 regulation. The *sin* mutation is present in iRhom2′s first transmembrane helix and results in reduced TNFα shedding [54]. It was later identified that the transmembrane helices of iRhom2 interact with ADAM17 [46,55], with the *sin* mutation disrupting that interaction and resulting in iRhom2′s inability to traffic ADAM17 efficiently out of the ER and into the Golgi [55]. The *sin* mutation also affected ADAM17′s substrate selectivity [55].

The same mechanism was later extended to ADAM17 shed EGFR ligands as well [53,56]. iRhom2 gain of function mutations in its N-terminal cytoplasmic tail, identified in the inherited Tylosis with oesophageal cancer (TOC) syndrome, result in increased EGFR ligand shedding and subsequent activation of EGFR signaling (Box 1) [57,58,59]. The same TOC gain of function mutations were also found to enhance shedding of TNFRs [60]. Consistently, two studies employing the same mouse curly bare (*cub*) iRhom2 mutation which deletes most of the cytoplasmic domain showed alterations in EGFR signaling [53,56]. Mechanistically, iRhom2 mediates ADAM17 trafficking and its subsequent activation which results in enhanced ADAM17 shedding activity (Figure 2) [45,46,47,49,61,62,63,64,65].

iRhom2 was found to efficiently bind mature ADAM17, and over-expressed iRhoms were found to localize to the plasma membrane, indicating a continued relationship at the cell surface [43,60,66]. Furthermore, ADAM17 substrate shedding differs depending on which of the two iRhoms is present [45]. This selectivity was recently shown to be regulated through direct iRhom-ADAM17 interaction via both their transmembrane helices and juxta-membrane domains, confirming their continued interaction at the cell surface [46]. However, for ADAM17 to actually shed its target substrates it needs to be at least momentarily relieved from iRhom binding [66,67]. In a similar fashion to ADAM17 regulation iRhoms are also regulated by post-translational modifications and protein-protein interactions, adding a further layer of control to the iRhom-ADAM17 relationship. ERK and MAPK-dependent phosphorylation of the cytoplasmic tail of iRhom2 results in 14-3-3 proteins binding to iRhom2 and its dissociation from ADAM17, which in turn allows stimulated shedding of ADAM17 targets [66,67]. Furthermore, and similar to control of ADAM17 activity through its cytoplasmic tail, this phosphorylation event has no effect on iRhom2 mediated ADAM17 maturation, and removal of the cytoplasmic tail altogether actually increased constitutive ADAM17 shedding activity of TNFR [60]. This indicates that direct cell surface phosphorylation controls iRhom-ADAM17 stimulated shedding. Further cementing the combined control of both proteins on the cell surface through their cytoplasmic tails, iRhom2 cytoplasmic tail binding partners also have a considerable effect on ADAM17 activity. Two groups published simultaneously that the four-point-one, ezrin, radixin, moesin (FERM) domain-containing protein 8 (FRMD8), also named the iRhom Tail-Associated Protein (iTAP), stabilizes the iRhom-ADAM17 complex at the cell surface. In its absence ADAM17 shedding of TNFα and EGFR ligands is impaired and the iRhom-ADAM17 complex is endocytosed and degraded in the lysosome [68,69].

As iRhom2 regulates EGFR ligand shedding, one would expect that the iRhom2 null mice share a similar open eye phenotype seen in EGFR and ADAM17 null mice [18,70]. This is not the case despite iRhom2 being expressed in the skin. iRhom2 deficient mice appear healthy, viable, and show no gross phenotype, while ADAM17 deficient mice exhibit severe symptoms after birth [18]. However, iRhom 1 and 2 double knockout mice show similar phenotypes compared to ADAM17 and EGFR null phenotypes, indicating redundancy in the cellular functions of the two iRhoms [48,49]. Indeed, iRhom1 triggers shedding of a selection of ADAM17 substrates in mouse embryonic fibroblasts [48]. Moreover, tissue- or cell type- specific expression of iRhom1 and iRhom2 might compensate for loss of one iRhom protein. iRhom1 seems to be important in neuronal tissue, excluding microglia [8,48,71,72,73], while iRhom2 may have more of a functional role in immune cells including brain microglia and liver hepatic stellate cells [48,52,54,61]. 

## 3. iRhom2 Regulated Pathways

### 3.1. TNF Signaling

TNFα is the founding member of a superfamily of cytokine-like molecules. Along with their cognate receptors, the TNF receptor (TNFR) superfamily, they control survival, proliferation and pro-inflammatory functions on both immune and non-immune cells [74]. TNFα binding to TNFR1 results in formation of two distinct complexes with diverging functions based on the cellular context; complexes I and II. Complex I consists primarily of the receptor itself, the adaptor protein TNFR1-associated death domain (TRADD), TNFR-associated factor 2 (TRAF-2), cellular inhibitors of apoptosis 1 and 2 (cIAP1/2), linear ubiquitin chain assembly complex (LUBAC) and receptor-interacting serine/threonine-protein kinase 1 (RIPK1). Based on the phosphorylation and ubiquitination status of RIPK1, this complex can activate downstream NF-κB signaling resulting in the expression of pro-inflammatory and pro-survival genes [75,76,77,78]. However, if NF-κB dependent transcription is inhibited or RIPK1 phosphorylation and ubiquitination status is changed this will alternatively result in formation of complex II [79,80]. Complex II, which is actually 3 distinct sub complexes a-c, results in cell death. Complexes IIa and IIb induce apoptosis, are caspase 8 dependent and contain either TRADD or RIPK1 respectively, while complex IIc, also known as the necrosome, depends on RIPK1, 3 and mixed lineage kinase domain-like (MLKL) to induce necroptosis [74,75,81].

TNFR1 is ubiquitously expressed and can be activated by both shed and membrane bound TNFα. TNFR2 on the other hand is expressed in a more limited fashion on immune, neuronal, cardiac, endothelial and stem cells binding membrane bound TNFα with higher affinity [74]. TNFR2 does not contain a death domain like TNFR1, however its signaling can result in cell death by loss of TRAF2. Mostly its activation induces NF-κB signaling resulting in more tissue regeneration and repair [74,82,83]. TNFα and both of TNFR 1 & 2 are shed by ADAM17 under regulation of iRhom2 [15,16,43,44,45,60,84]. However, the effect of iRhom2 on shedding of ADAM17 ligands—specifically, TNFα-versus TNFR shedding—can have opposing consequences. The outcome is context specific and depends on the cellular composition and pathophysiological mechanisms within each tissue (Figure 3).

iRhom2 null mice and those with the *sin* mutation, that inhibits iRhom2′s ability to efficiently traffic ADAM17 from the ER to the Golgi, have reduced soluble TNFα (Table 1) [43,44,54]. The TNFα-TNFR1 pathway is crucial for bacterial defense [85,86,87,88]. Mice with reduced soluble TNFα due to iRhom2 absence succumbed to sublethal doses of *Listeria monocytogenes* [44]. They were also more likely to develop severe disease after myocardial infarction due to altered TNFα signaling and macrophage polarization [89], and were more likely to develop atherosclerosis when challenged with a high fat diet, although the effect was not solely due to altered TNFα [90]. However, this is at odds with a report of increased TACE expression and TNFα in localized samples of ruptured plaques in human acute myocardial infarction, which also indicates a more complex process beyond TNFα shedding alone [91]. Furthermore, iRhom2 null mice are more likely to develop fibrosis post bile duct ligation, as a model of human cholestatic disease, due to reduced TNFR1 shedding from the surface of hepatic stellate cells and subsequent enhanced TNF signaling [52]. The same mice conversely are protected from LPS induced septic shock [44], rheumatoid arthritis [61,92], hemophilic arthropathy [93], lupus nephritis [64], post intestinal ischaemia reperfusion acute lung injury [94], renal injury from particulate matter [95] and inflammatory bowel disease [96], all of which are TNFα mediated with the exception of lupus nephritis which is TNFα and HB-EGF mediated [64]. A previous study shows the iRhom2 null mice being more susceptible to inflammatory bowel disease using a spontaneous colitis mouse model relating it to altered T helper cell cytokine production [97].

Accordingly, there may likely be more nuances to iRhom2′s role in inflammatory bowel disease as yet undiscovered, especially that iRhom2 T cell modulation has been reported elsewhere. Adoptive transfer of iRhom2 null CD8^+^ T cells into CD8 null mice with an ongoing viral infection resulted in a greater expansion of the immunosuppressive Vβ5+ regulatory T cells (Tregs), indicative of increased membrane TNFα-TNFR2 signaling [106,107,108,109]. iRhom2 null mice do indeed have increased presence of unprocessed membrane bound TNFα on their CD8^+^ T cells [106]. Conversely, increased iRhom2 expression in chronic inflammation results in enhanced membrane TNFα cleavage from endothelial cells. This results in reduced membrane TNFα-TNFR2 signaling to the detriment of the healing process [62]. Finally, increased TNFR shedding under inflammatory conditions, in response to increased iRhom2 and TACE phosphorylation, protects hepatocytes through reduced TNFR1 signaling and soluble TNFRs binding circulating TNFα [84,110]. Hence, depending on the cellular tissue composition in the disease process and the equilibrium of TNFα shedding vs its receptors, the outcome of iRhom2 regulation of TNF signaling varies greatly.

### 3.2. EGFR Signaling

EGFR, also known as ErbB-1, is a member of a family of four transmembrane tyrosine kinase receptors. It is activated through binding of its ligands to the extracellular domain, which leads to dimerization, trans-autophosphorylation, internalization and either recycling or degradation of the receptors dependent on ligand binding affinity and duration. This results in ligand specific activation of genes responsible for cell proliferation, migration, survival, and differentiation [115]. Signaling pathways activated by EGFR include MAPK, PI3K/AKT, JAK/STAT, and PKC [115,116,117,118]. The EGFR ligands can be membrane bound and shed by ADAM10 and ADAM17 [115,119]. Of the seven ligands, amphiregulin (AR), heparin-binding-EGF-like growth factor (HB-EGF), epiregulin (EPR), TGFα and epigen have been reported to be shed by ADAM17 [115,119,120]. EGFR signaling is required for normal epithelial development and homeostasis, as is evidenced by several mouse knock out models. In addition to the open eye phenotype, EGFR deficient mice exhibit developmental defects in several organs including skin, lung and gastrointestinal tract [70]. Furthermore, several epithelial cancers are characterized by enhanced EGFR activation [115,121,122,123,124]. This correlates with a poor response to conventional therapy [125,126]. Increased EGFR signaling in epithelial cancers occurs by a number of mechanisms, including over-expression of EGFR [126,127,128] or its ligands [27,117,121], defective ligand processing [129], activating mutations in EGFR [130,131], or excessive EGFR transactivation [132,133,134,135,136]. 

There has been a large body of work done relating iRhoms to EGFR ligand shedding and their role in several EGFR related diseases. Autosomal-dominant inherited TOC familial cancer syndrome patients were then found to have an activating mutation in iRhom2 that increased EGFR ligand shedding, firmly putting iRhom2 in the midst of EGFR related cancers as well as epithelial homeostasis (Box 1) [57,58,59]. The syndrome is characterized by palmar and plantar hyperkeratosis, accelerated wound healing, oral leukoplakia, and a markedly elevated risk of developing esophageal squamous cell carcinoma [58,98,99,100,101,102,103]. In mouse models of the syndrome it was shown that deletion of the EGFR ligand AR or ADAM17 were able to restore normal skin phenotype [137,138] and that the defect was skin specific [139]. While this cancer syndrome is extremely rare it gave us important insights into the iRhom2-EGFR relationship. This relationship in squamous cancers was further cemented when hepatocyte growth factor (HGF) signaling through its receptor MET was found to be dependent on both iRhom1 and 2 regulation of ADAM17 shedding of EGFR ligands [47]. Indeed, iRhom1 had already been reported to regulate EGFR ligand release and EGFR transactivation in vitro [140]. Recently both iRhoms have been shown to play a role in cervical cancer. Cervical cancer tissue from 83 patients was found to have high expression of both iRhom 1 and 2 compared to normal tissue. High expression was correlated with poor outcome, which the authors relate through in vitro data to EGFR, WNT/β-Catenin and TGFβ signaling [65]. Carcinogenesis however is a process that involves more than one pathway as is evidenced by the mixed results of tyrosine kinase inhibitor clinical trials and increased dual pathway inhibition approaches [141,142]. Furthermore, EGFR signaling is involved in much more than epithelial cancers alone. As mentioned above, lupus nephritis is dependent on both TNF and EGFR signaling. In a mouse model of lupus nephritis, lack of iRhom2 resulted in protection from progressive renal injury via these pathways [64]. 

### 3.3. Notch Signaling

The notch receptor is cleaved by ADAM 10 and 17. However, cleavage can only occur in the presence of a bound ligand that induces a conformational change in the receptor. Alternatively, mutations in the negative regulatory domain (NRR) of the notch receptor relieve this requirement [143,144]. Once cleaved, the remaining transmembrane and cytoplasmic portions are then processed by γ–secretase releasing the notch intracellular domain [145], which translocates to the nucleus affecting transcription [146]. The notch pathway is active during development and afterwards. It is responsible for regulating replication, differentiation and the maintenance of stem cells [147]. It has a prominent role in tumorigenesis acting as both a tumor suppressor in squamous cell carcinoma [148], and an oncogene in T-cell leukemia [149]. This is the result of its transcriptional regulation being heavily dependent on the cellular epigenetic context [147]. There are 4 notch receptors in humans (NOTCH1-4). The NOTCH1 NRR is commonly mutated in T-cell leukemias and its ligand-independent cleavage is ADAM17 dependent [143], signifying a potential role for iRhom2 in notch dependent carcinogenesis.

There are currently two studies relating iRhom2 to notch signaling regulation via ADAM17 receptor cleavage. One relates the role of iRhom2 to hepatocellular carcinoma (HCC) development, specifically in liver cancer stem cells [111]. In 90 human HCC samples the authors found that samples which had increased expression of stem cell markers also had elevated inducible nitric oxide synthase (iNOS), active ADAM17 and notch signaling, all of which were iRhom2 dependent [111]. While increased iNOS activity has been shown to induce ADAM17 activity and be liver protective through cleavage of TNFR1 [84], in this case it was associated with enhanced notch signaling and poor survival [111]. The second paper relates to the spontaneous iRhom2 uncovered (*Uncv*) mouse mutation which displays a hairless phenotype in the BALB/c background [112]. In these mice iRhom2 is incapable of supporting ADAM17 maturation which leads to reduced notch receptor processing and reduction in hair shaft cell proliferation and development [112,113]. 

### 3.4. MAVS & STING

The innate immune response to non-self molecules (e.g., microbial products), known as pathogen associated molecular patterns (PAMP), induces transcription of pro-inflammatory genes required to control infections and mount an effective adaptive immune response [150]. PAMPs are recognized through various germ-line inherited pattern recognition receptors (PRR) that induce production of pro-inflammatory cytokines [151]. Viral PAMPs include their genome, especially in ways differentiating them from host DNA or RNA [152]. Viral RNA is recognized through the retinoic acid-inducible gene-I (RIG-I) like receptor pathways via mitochondrial antiviral-signaling protein (MAVS, also known as VISA, IPS-1 and Cardif) [153,154,155], while viral DNA is recognized through the cyclic GMP-AMP synthase (cGAS) via stimulator of interferon genes (STING) pathway [156]. 

Once RIG-I binds viral RNA, a conformational change has been suggested allowing it to bind to MAVS [155]. However, it was shown without over-expression that RIG-I binds to MAVS even in the absence of a viral infection, although no interferon regulatory transcription factor 3 (IRF-3) recruitment to the complex occurred [157]. Regardless of how the interaction is induced, once the complex is complete, innate immune antiviral responses can ensue via activation of IRF-3 and NF-κB signaling [157,158]. MAVS can be sent off for proteasomal mediated degradation by several E3 ubiquitin ligases including the E3 ligases Ring Finger Protein 5 (RNF5) and membrane-associated ring finger (C3HC4) 5 (MARCH5) [159,160]. In the absence of iRhom2, murine innate immunity to Vesicular Stomatitis Virus (VSV) infection was considerably impaired leading to neurological symptoms in all iRhom2 deficient mice compared to 60% of wild-type mice. The authors also found that absence of iRhom2 resulted in reduced MAVS levels [10]. Mechanistically, iRhom2 was reported to induce auto-ubiquitination of the E3 ligase RNF5 either in the absence of viral infection or during early viral infections (4h). This resulted in proteasomal mediated degradation of RNF5 rescuing MAVS itself from proteasomal mediated degradation. During late infections (>8h) iRhom2 reduced MARCH5 levels saving MAVS again from proteasomal mediated degradation [10]. 

In response to detecting viral DNA, cGAS produces the second messenger cGAMP, which in turn binds to STING in the ER. STING is then translocated in an iRhom2 dependent manner from the ER to the Golgi and on to the perinuclear microsomes, in a mechanism reminiscent of iRhom-ADAM17 trafficking, albeit mediated by iRhom2′s recruitment of the translocon-associated protein (TRAPβ) [9]. However, iRhom2 also stabilizes STING by recruiting a deubiquitylating enzyme that rescues STING from proteasomal mediated degradation. iRhom2 deficient mice were thus more susceptible to lethal herpes simplex Virus 1 (HSV-1) infections than their wild type counterparts [9]. It further seems that this is a conserved mechanism, as other DNA viruses have evolved mechanisms to evade the innate immune response by disrupting the iRhom2-STING interaction to induce viral latency [114]. Human cytomegalovirus (hCMV) encodes the tegument protein UL82 which is capable of inhibiting the STING-iRhom2-TRAPβ translocation complex. Moreover, in its absence antiviral gene responses downstream of STING were enhanced [114]. 

### 3.5. K6/16 Balance

iRhom2 has a central role in the release of EGFR ligands. An activating mutation of iRhom2, identified in TOC patients, results in palmar and plantar hyperkeratosis, accelerated wound healing, oral leukoplakia, and a markedly elevated risk of developing esophageal squamous cell carcinoma. In addition to iRhom2-induced ADAM17 EGFR ligand processing, iRhom2 can bind directly to the stress-associated keratin K16 playing a pivotal role in the skin characteristics of the syndrome [104]. Keratins are thought to be mostly responsible for skin resilience of the palms and soles as they constitute a vast majority of the proteins expressed in keratinocytes [161,162,163]. Keratins are expressed in pairs of acidic and basic keratins which heterodimerize. For example, the acidic K16 keratin is paired with the basic K6 keratin [164]. K16 is expressed in palmoplantar epidermis [164], at sites of wound healing [165], associated with squamous cell carcinomas and induced during inflammation [166]. As K16 mutations also present with palmar and plantar hyperkeratosis [167], the authors investigated iRhom2 gain of function mutations in TOC and their relation to K16 expression. They find in hyperproliferative TOC keratinocytes an enhanced iRhom2 K16 interaction, in the same binding region used for K6/K16 dimerization. This altered K16 filament organization and reduced expression of its binding partner K6. While in the absence of iRhom2, proliferation and K16 expression are reduced and associated with thinning of the epidermis of the mouse footpad. Changes in K6 and K16 expression are found on the transcriptional level [104]. While iRhom2 activity was already linked to squamous cell carcinoma of the esophagus in TOC patients, new evidence further links iRhom2 expression to oral squamous cell carcinomas [105]. iRhom2 over-expression was correlated with reduced patient survival and enhanced migration of an oral squamous cell carcinoma line in vitro [105]. In light of the effects of iRhom2 on K6/16 balance in skin keratinocytes as well as its role in EGFR signaling, further investigation is required on iRhom2′s role in wound healing and related squamous carcinomas.

Box 1iRhom2 in human diseases.Tylosis with Oesophageal Cancer (TOC) is an autosomal
dominant syndrome characterized by palmar and plantar hyperkeratosis,
accelerated wound healing, oral leukoplakia, and a markedly elevated risk of
developing esophageal squamous cell carcinoma. Using family pedigrees and DNA
samples from 3 US, UK and German families, single amino acid substitutions in
exon 6 of RHBDF2 were identified as the underlying cause of the syndrome [58]. This was further confirmed independently in
a Finnish family [59] and an African family [103] showing again a single amino acid
substitution between the two identified before and all within the highly
conserved region of the N-terminal cytoplasmic tail of iRhom2: a p.Ile186Thr
mutation in the US and UK families, a p.Asp188Asn mutation in the Finnish
family, a p.Asp188Tyr in the African family and a p.Pro189Leu mutation in the
German family. These mutations lead to an altered distribution of iRhom2 in
skin and dysregulated EGFR signaling [58].
The underlying mechanism was related to the mutations activating iRhom2 which
leads to increased ADAM17 maturation and activity in epidermal keratinocytes
from TOC patients, which in turn increases shedding of TNFα, AR, TGFα and
HB-EGF and enhances EGFR phosphorylation [57].
Consistently, both iRhom1 and iRhom2 expression was found to be enhanced in
all histological cervical carcinoma types. iRhom1, iRhom2 and Ki-67
expression were found to correlate with increased tumor stage, invasion and
poor clinical outcome [65]. Furthermore,
oral squamous cell carcinoma patient samples have been shown to over-express
iRhom2, which was found to correlate with poor patient survival but no other
clinico-pathological variables [105].
Moreover, iRhom2 expression was increased in cirrhotic liver samples [52], kidney tissue of Lupus nephritis patients [64], and colon samples from inflammatory bowel
disease patients [96]. 

## 4. Discussion & Outlook

Taken together, iRhom2 has multiple functions during infections and diseases. iRhom2 was originally linked to ADAM17 maturation and activation and accordingly to functions related to ADAM17. However, its role during DNA and RNA viral defense through MAVS and STING suggests that iRhom2 might have a central role during innate immune functions. Consistently, iRhom2 is associated with several pathological conditions such as autoimmunity and cancer. Notably, severe phenotypes observed in ADAM17 deficient mice are not observed in mice deficient for iRhom2 alone, which might suggest little side effects for a therapy targeting iRhom2 specifically. Further studies could focus on the role of iRhom2 in other ADAM17 substrates Moreover, a more thorough characterization of both, iRhom1 and iRhom2 in multiple model systems will help to further clarify their specific roles in these and other pathways in health and disease. 

## Figures and Tables

**Figure 1 ijms-21-06570-f001:**
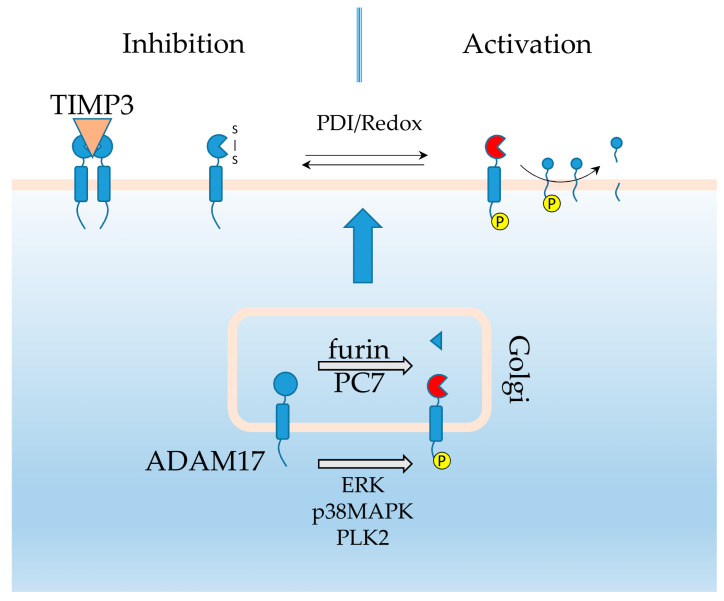
ADAM17 post translational regulation. ADAM17 is processed in the Golgi to remove its inhibitory pro-domain. Its cytoplasmic tail is phosphorylated, but that is not required for its transport from the Golgi to the cell surface. However, when not phosphorylated this induces ADAM17 dimerization and TIMP3 binding resulting in inactivation. Alteration in the disulfide bridge arrangements can also alter ADAM17′s activation status. The phosphorylation status of the substrates themselves can also alter ADAM17 shedding performance.

**Figure 2 ijms-21-06570-f002:**
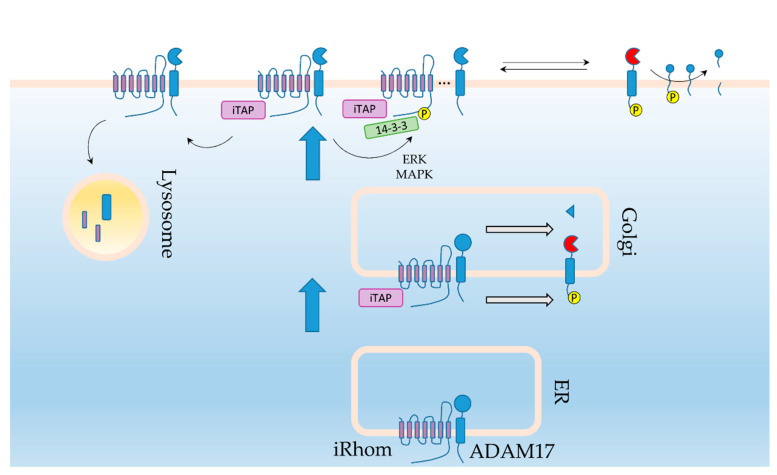
iRhom regulation of ADAM17. iRhom binds to ADAM17 in the ER facilitating its export to the Golgi where it can begin its posttranslational modification fueled activation. iRhom and ADAM17 continue to the cell surface where in the presence of iTAP the complex is stabilized. In its absence the complex is sent for endocytosis and lysosomal degradation. Phosphorylation of iRhom by ERK and MAPK allows iRhom to bind to 14-3-3 proteins, which in turn facilitate release of ADAM17 from the complex to allow final activation and shedding of its targets.

**Figure 3 ijms-21-06570-f003:**
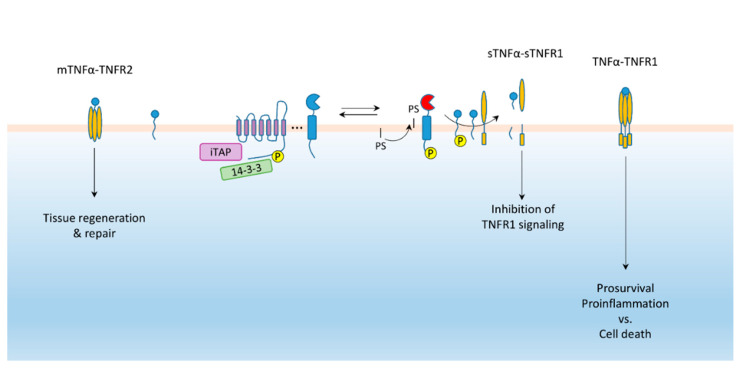
iRhom-ADAM17 regulation of TNF signaling. TNFR1 signaling depends on whether TNFα or TNFR1 is predominantly cleaved. In the absence of TNFα cleavage TNFR2 signaling is favored.

**Table 1 ijms-21-06570-t001:** iRhom2 regulated pathways in immunity and disease.

Pathway	Immune process/ Disease	Mechanism	References
**EGFR**	Tylosis with oesophageal cancer	Activating iRhom2 mutation, increased EGFR ligand shedding	[57,58,59,98,99,100,101,102,103]
	Increased squamous cell carcinoma	iRhom expression induction, increased EGFR ligand shedding, increased MET signaling	[47]
	Increased cervical cancer	Increased iRhom expression, increased EGFR, WNT and TGFβ signaling	[65]
	Lupus nephritis resistance	Reduced iRhom2, soluble TNFα and HB-EGF	[64]
**K6/16**	Tylosis with oesophageal cancer & keratinocyte skin homeostasis	TOC mutated iRhom2 binds K16 altering K6-K16 dimerization. Reduced iRhom2, reduced K16 expression.	[104]
	Increased oral squamous cell carcninoma	Increased iRhom2 expression, migration and proliferation	[105]
**TNF**	Resistance to LPS induced septic shock	Reduced iRhom2,soluble TNFα and TNFR1 signaling	[43,44]
	Bacterial defense defect	Reduced iRhom2,soluble TNFα and TNFR1 signaling	[44]
	Chronic inflammation & reduced healing	Induction of iRhom2 expression resulting in reduced membrane TNFα-TNFR2 signaling	[62]
	Rheumatoid arthritis resistance	Reduced iRhom2 and soluble TNFα	[61,92]
	Increased regulatory T cell expansion	Reduced iRhom2, increased membrane TNFα-TNFR2 signaling	[106,107,108,109]
	Inflammation & hepatoprotection	Induction of iRhom2 expression, increased TNFR shedding, reduced TNFR1 signaling	[84,110]
	Worsened atherosclerosis & myocardial infarction outcomes	Reduced iRhom2, altered TNF signaling and macrophage polarization	[89,90]
	Hemophilic arthropathy resistance	Reduced iRhom2 and soluble TNFα	[93]
	Lupus nephritis resistance	Reduced iRhom2, soluble TNFα and HB-EGF	[64]
	Intestinal ischaemia reperfusion acute lung injury resistance	Reduced iRhom2 and soluble TNFα	[94]
	Particulate matter Renal injury resistance	Reduced iRhom2 and soluble TNFα	[95]
	Inflammatory bowel disease resistance	Reduced iRhom2 and soluble TNFα	[96]
	Inflammatory bowel disease susceptibility	Reduced iRhom2 and IL10 resulting in altered T helper cell cytokine production	[97]
	Increased cholestatic liver fibrosis	Reduced iRhom2, reduced TNFR1 shedding in hepatic stellate cells and increased TNFR1 signaling	[52]
**Notch signaling**	Hepatocellular carcinoma induction	Increased iRhom2 activity by inducible nitric oxide synthase, ADAM17 cleavage of Notch receptor	[111]
	Reduced hair development	Spontaneous iRhom2 mutation, reduced ADAM17 activity and Notch receptor processing	[112,113]
**MAVS & STING**	Defective innate immune response to RNA viruses	Reduced iRhom2, increased E3 ubiquitin ligase, reduced MAVS by increased proteasomal degradation	[10]
	Defective innate immune response to DNA viruses	Reduced iRhom2, reduced STING nuclear translocation & increased proteasomal degradation	[9,114]

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
