# Peer review of "iRhom2: An Emerging Adaptor Regulating Immunity and Disease"

_ijms, 2020, doi:10.3390/ijms21186570_

Round 1

Reviewer 1 Report

In their review “iRhom2: An emerging adaptor regulating immunity and disease.” Dr. Mazin Al-Salihi and Dr. Philipp Lang give a comprehensive overview of the current state of research regarding the biology of the pseudoprotease iRhom2. iRhom2 is an important adapter protein for central hubs of different crucial (patho-)physiological signals such as ADAM17- and Sting-dependent pathways. This review points out (recent) developments which give insight into functions and putative molecular mechanisms of iRhom2 and gives a thorough overview of its (patho-)physiological importance especially regarding TNF signalling, EGFR signalling and virus detection.

Overall, I have only some minor comments:

  1. There are a few typos which need to be corrected, e.g.: p.7 l.255 “the us” is not needed in the sentence, p.9 l.33 “that” should be “than”
  2. It should be avoided to refer to transmembrane helices as transmembrane domains, since domains are defined as tertiary structures containing several secondary structure subunits. Hence, single transmembrane helices do not qualify as domain. In case of rhomboids such as iRhom2 six or seven transmembrane helices form the rhomboid-like domain.
  3. 2 l.48 EGF is not an ADAM17 substrate; the authors most likely intended to write “HB-EGF” or “TGF”
  4. 2 l.84 iRhom1 should also be mentioned as it is functionally similar to iRhom2 in terms of ADAM17 maturation, forward trafficking, ...

Author Response

In their review “iRhom2: An emerging adaptor regulating immunity and disease.” Dr. Mazin Al-Salihi and Dr. Philipp Lang give a comprehensive overview of the current state of research regarding the biology of the pseudoprotease iRhom2. iRhom2 is an important adapter protein for central hubs of different crucial (patho-)physiological signals such as ADAM17- and Sting-dependent pathways. This review points out (recent) developments which give insight into functions and putative molecular mechanisms of iRhom2 and gives a thorough overview of its (patho-)physiological importance especially regarding TNF signalling, EGFR signalling and virus detection.

Thank you for your in-depth delve into our review and your comments to improve our manuscript.

Overall, I have only some minor comments:

There are a few typos which need to be corrected, e.g.: p.7 l.255 “the us” is not needed in the sentence, p.9 l.33 “that” should be “than”

Thank you for pointing this out, we corrected as suggested:

L298: "the us" corrected to "us"

L385: "that" corrected to "than"

It should be avoided to refer to transmembrane helices as transmembrane domains, since domains are defined as tertiary structures containing several secondary structure subunits. Hence, single transmembrane helices do not qualify as domain. In case of rhomboids such as iRhom2 six or seven transmembrane helices form the rhomboid-like domain.

This is an important point. We now refer to: iRhom2 transmembrane helices in L129-130, L155.

2 l.48 EGF is not an ADAM17 substrate; the authors most likely intended to write “HB-EGF” or “TGF”

Thank you for pointing to this important point. We now write TGFα in accordance to the papers cited (L56-57).

2 l.84 iRhom1 should also be mentioned as it is functionally similar to iRhom2 in terms of ADAM17 maturation, forward trafficking, ...

 L112: iRhom1 is now mentioned along with iRhom2 in terms of ADAM17 maturation and the references updated accordingly.

Reviewer 2 Report

The manuscript by Al-Sahihi and Lang reviews the pseudoprotease iRhom2 as a regulator of immunity and disease. However, the topic is extensively reviewed recently by many groups (for example; Geesala et al 2019; Dusterhoft et al 2019; Dulloo et al 2019; Ticha et al 2018; etc …), therefore the novelty of this work is questioned. Although the role of iRhom2 in disease is claimed to be the highlight of this manuscript, this information is buried in long paragraphs describing iRhom2 molecular mechanisms (maybe introducing tables would help).

The manuscript needs major changes to organise the flow of information. For example, the authors reviewed ADAM17 post-translational modifications in details, which is not the objective of their review, as well as discussing iRhom in Drosophila in the context of human disease (lines 244-248).

The manuscript needs extensive editing and fact-checking as numerous ambiguous sentences (lines 27, 54, 270, 357 etc …), spelling mistakes, and incorrect information were detected (for example, Line 229: the references cited by the authors (and many others) refer to 7 ligands not 8). Abbreviations should be spelled out when first mentioned, and the abbreviations should be used thereafter. The conclusion section should include the authors view on the topic, not to introduce more data on iRhom2-regulated substrates (i.e. L-selectin, IL-6R and trans-signaling).

Author Response

The manuscript by Al-Sahihi and Lang reviews the pseudoprotease iRhom2 as a regulator of immunity and disease. However, the topic is extensively reviewed recently by many groups (for example; Geesala et al 2019; Dusterhoft et al 2019; Dulloo et al 2019; Ticha et al 2018; etc …), therefore the novelty of this work is questioned.

We thank the reviewer for the time they afforded us in reviewing our paper. We now refer to these reviews and cite them (L44-45). Since iRhom2 is an emerging adaptor with new manuscripts being published, we also discuss the recent literature. We have also taken a different approach to the previous reviews by organizing the reviewed knowledge based on signaling pathways. We have now contrasted the approach to the subject along with the mentioned reviews.

Although the role of iRhom2 in disease is claimed to be the highlight of this manuscript, this information is buried in long paragraphs describing iRhom2 molecular mechanisms (maybe introducing tables would help).

This is an important point. We have now introduced a table summarizing the immune processes and diseases reviewed in the paper as well as a Box specifically outlining the human diseases where iRhom2 has been linked in human samples (new Table 1 and Box1)

The manuscript needs major changes to organise the flow of information. For example, the authors reviewed ADAM17 post-translational modifications in details, which is not the objective of their review,

as well as discussing iRhom in Drosophila in the context of human disease (lines 244-248).

We have revised the section of iRhom2 control of ADAM17 and now only briefly mention the modifications of ADAM17, which are related to iRhom2 (L50-192). For example, the same kinases phosphorylating ADAM17 are responsible for 14-3-3-dependent iRhom release from ADAM17 to allow shedding. However, if the reviewer feels like the section on the posttranslational modification of ADAM17 should be removed from the manuscript we would be happy to do so. Moreover, we removed the statement regarding the role of iRhom in Drosophila and highlight the clinical significance instead. (L287-291).

The manuscript needs extensive editing and fact-checking as numerous ambiguous sentences (lines 27, 54, 270, 357 etc …), spelling mistakes, and incorrect information were detected (for example, Line 229: the references cited by the authors (and many others) refer to 7 ligands not 8).

These are important points.

We have rechecked the references cited for accuracy throughout and fixed ambiguous sentences (L27, L50-52, L60, L79-81, L315-319, L397-417). We have corrected several spelling mistakes. Moreover, we now refer to 7 ligands as described in references '95 and '96 (L270-271).

Abbreviations should be spelled out when first mentioned, and the abbreviations should be used thereafter.

We now spell out abbreviations (L14-15).

 The conclusion section should include the authors view on the topic, not to introduce more data on iRhom2-regulated substrates (i.e. L-selectin, IL-6R and trans-signaling).

We have adjusted the section to better reflect our opinion of iRhom2 as a central player in both immunity and disease and its potential as a low side effect target for therapeutic intervention and removed the other statements (L419-442).

Round 2

Reviewer 2 Report

None, accept as is.